# Elemental Composition of Plankton Exometabolites (Mucous Macroaggregates): Control by Biogenic and Lithogenic Components

**DOI:** 10.3390/metabo13060726

**Published:** 2023-06-05

**Authors:** Nives Kovač, Jérôme Viers, Jadran Faganeli, Oliver Bajt, Oleg S. Pokrovsky

**Affiliations:** 1Marine Biological Station, National Institute of Biology, Fornače 41, SI-6330 Piran, Slovenia; 2Geosciences and Environment Toulouse, UMR 5563 CNRS, 14 Avenue Edouard Belin, 31400 Toulouse, France; 3BIO-GEO-CLIM Laboratory, Tomsk State University, 634050 Tomsk, Russia; 4Institute of Ecological Problems of the North, N. Laverov Federal Center for Integrated Arctic Research, Russian Academy of Sciences, Nab. Severnoi Dviny 23, 163000 Arkhangelsk, Russia

**Keywords:** mucilage, exopolysaccharides, macroaggregates, exometabolites, northern Adriatic, elemental composition

## Abstract

Among the various exometabolitic effects of marine microorganisms, massive mucilage events in the coastal zones of temperate and tropical seas are the most spectacular and environmentally important. Abundant mucilage material in the form of aggregates appears in late spring/early summer in the water column of the Adriatic Sea. These macroaggregate biopolymers originate mainly from plankton exometabolites, with both autochthonous and allochthonous components, and strongly impact the tourism, fisheries, and economy of coastal countries. In contrast to extensive studies on the structural and chemical nature of macroaggregates performed over past decades, the full elemental composition of these substances remains poorly known, which does not allow for a complete understanding of their origin, evolution, and necessary remediation measures. Here, we report the results of comprehensive analyses of 55 major and trace elements in the composition of macro aggregates collected at the surface and in the water column during massive mucilage events. Through normalization of the elemental chemical composition of the upper earth crust (UCC), river suspended material (RSM), mean oceanic plankton, and mean oceanic particulate suspended material, we demonstrate that the water column macroaggregates reflect a superposition of the signal from plankton and marine particulate matter. The surface macroaggregates were preferentially enriched in lithogenic component, and carried the signature of planktonic material. The rare earth element (REE) signal was strongly dominated by plankton and, to a lesser degree, by oceanic particulate matter, while at the same time being strongly (>80 times) impoverished compared with UCC and RSM. Taken together, the elemental composition of macroaggregates allows for distinguishing the lithogenic and biogenic impacts on the occurrence of these unique large-scale mucilage events, linked to the exometabolism of marine plankton combined with the input of allochthonous inorganic material.

## 1. Introduction

The importance of the element interaction with organic matter in seawater stems from the fact that this is the main governing factor of trace elements’ geochemical behavior and their bioavailability. The majority of seawater organic matter originates from phytoplankton activity, in the form of either cellular biomass or soluble and insoluble exometabolites. Among various organic metabolites, macroscopic mucilage is among the most spectacular and still poorly understood phenomena. The episodic mucilage events of the northern Adriatic are represented by a hyper-production of mucilage material in the form of aggregates of varying sizes and shapes located at the surface, in the water column and attached to the sea bottom. They normally appear in the late spring/early summer and their presence could be observed from one to several months, i.e., from late May/June to September [1]. At the same time, massive production of plankton exometabolites in the Northern Adriatic does not happen each year, and it still remains an unpredictable event. The persistence/resistance of macroaggregates in the summer stratified northern Adriatic water column is related to their complex structure and mineral–organic matter interactions [2]. They represent an important site of accumulation, transformation, and degradation of organic matter, contributing to the transport, distribution, and fate of particulate matter in seawater. Besides an important environmental role, they exhibit a serious impact on the tourism, mariculture, fisheries, and economy of coastal countries. Once formed, macroaggregate biopolymers are subjected mainly to bacterial degradation, as well as to various chemical and photochemical transformations [3,4]. At the end of the summer season, rain storm events, change in the water column hydrographic structure, and modification of the water motion/circulation pattern [5,6] usually lead to the disappearance of mucous macroaggregates.

The major organic and inorganic compositions of northern Adriatic macroaggregates have been extensively studied ([1] and references therein), but a detailed elemental composition of mucilage is still missing. Considering the general agreement about the mucilage phytoplankton origin, including their exudates and lyzed matter [7,8,9], there is no doubt that the biotic component is a crucial factor determining the basic elemental signature of northern Adriatic macroaggregates. At the same time, macroaggregates are characterized by heterogeneous composition, comprising phytoplankton, bacteria and cyanobacteria, meso-zooplankton, micro-zooplankton, and zooplankton debris (i.e., crustacean cuticles and antennae and faecal pellets), yeasts, pollen and various inorganic components such as empty frustules of diatoms and skeletal remains of coccolithophorids, empty thecae of dinoflagellates, and mineral particles [1]. High-resolution electron microscopy demonstrated a very complex, honeycomb-like structure of the mucus macroaggregates that might grow to macroscopic sizes [10]. In accordance with their gel-like nature, macroaggregates can contain more than 90% water. The organic matter/organic carbon ratio ranges from 2 to 3.6 [11,12]. The organic C and total N contents varied from 5 to 35% and 0.5 to 4.4%, respectively. The Corg/N atomic ratios in mucous macroaggregates ranged from 5.8 to 28.7 [1,13]. About 30–60% of their organic fraction is represented by polysaccharides, proteins, and lipids [4,14,15] and, in a minor part, by aromatic compounds. Extensive spectroscopic studies demonstrated that the general structure of macroaggregates contains four major classes of elements: carbohydrates, ester and amide functional groups, aliphatic components, and organosilicon components [2,4,16,17,18,19]. The carbohydrate content in the mucilage was assessed via 3-methyl-2-benzothiazoline hydrazine hydrochloride (MBTH) and 2,4,6-tripyridyl-s-triazine (TPPZ) assays; characterization of oligosaccharides using HPLC/RI revealed maltose and pentaose as the main components [14]. The high molecular weight of the water-soluble fraction of mucous macroaggregates was confirmed by size exclusion chromatography (SEC). Four major classes of structural elements of macroaggregates were identified: carbohydrates, ester and amide functional groups, aliphatic components, and organosilicon components. The spectroscopic analyses showed the same general structural pattern of “fresh” and more aged macroaggregates samples, indicating the preservation of organic matter during the mucilage event [1]. These compositional and structural features contribute to the gel-like and sticky character of macroaggregates. In contrast, the inorganic component (20 to 80%) is mostly represented by scavenged and incorporated autochthonic and allochthonous particles [20]. A lower proportion of the inorganic fraction is observed in aggregates formed at the beginning of mucilage events, whereas the bottom and precipitated macroaggregate samples are preferentially enriched in mineral components [4,12,20].

The chemical and biological characteristics of northern Adriatic seawater are capable of affecting, directly and indirectly, the mucous elemental composition. Among the possible sources of major and trace elements are different industrial activities at the coastal zone (power and chemical plants), ports, urban areas, agriculture areas, and tourist areas located along the coast. Furthermore, this area is exposed to pollution from the inflow [21] of northern Italian rivers (mainly the Po River and smaller ones such as Isonzo, Taglimento, Livenza, Piave, Brenta, and Adige, including the local Slovenian rivers (Dragonja, Rižana). Finally, besides an important freshwater inflow, and given its shallowness, the northern Adriatic is subject to highly variable atmospheric forces, including significant seasonal African dust fallout [22]. During the mucilage events, such inputs (dry and wet deposition and river discharging) and meteorological and water column conditions play an important role in the elemental signature of macroaggregates.

The aim of this study was to investigate the multi-elemental composition of northern Adriatic macroaggregates, to compare it to average oceanic plankton and lithogenic sedimentary material, and to evaluate the differences in the element content of their water (interstitial) and gelatinous (matrix) fractions. Specifically, we hypothesized that (1) the elemental composition of macroaggregates will be different depending on their location, i.e., floating on the water surface or suspended within the water column, and (2) the major and trace element concentration in the solid phase can reflect a superposition of plankton exometabolites (macro- and micronutrients linked to organic molecules), suspended-mineral-originated (silicate, carbonate), biologically indifferent, refractory trace element, atmospheric dust fallout (toxicants), and passive uptake of trace cations from the surrounding seawater. Although the rather small number of samples cannot allow for the quantification of the partial contribution of each source, this first study of multi-elemental composition of plankton exometabolites may provide a background for further, highly spatially and seasonally resolved research.

## 2. Materials and Methods

### 2.1. Mucilage Events

The mucilage event of 2000 started in the first week of June, with the maximal presence of mucous material at the surface and in the water column at around 10–22 June. The sedimentation of some portion of mucilage followed a mid-June peak, after which only sporadic mature macroaggregates were observed in the water column [1,10,13,20]. They were mostly accumulated at the bottom and observed as highly degraded/mineralized settled aggregates [1,15,16]. In the eastern part of the Gulf of Trieste, they disappeared by the end of July [23].

In 2004, the mucilage event starting in the middle of May was firstly observed in the form of marine snow and small aggregates within the water column, followed by more intense aggregation processes during June 2004, which firstly led to the appearance of large quantities of surface aggregates, and later to the formation of denser and more gel-like macroaggregates [1,13]. In the middle of July 2004, owing to the decline in mucilage production, the water column became transparent and a sizable amount organic matter accumulated in the bottom layers. In summer 2004, the macroaggregates were densely populated by a single species of dinoflagellates, *Gonyaulax fragilis*, at the beginning, and later became colonized by the typical “mucous community”, with predominating diatoms, flagellates, dinoflagellates, cyanobacteria, and heterotrophic bacteria [1,15]. In particular, the sampled mucilage events were accompanied by the development of three plankton groups: (a) diatoms: *Cylindrotheca closterium*, *Cyclotella* sp., *Pseudo-nitzschia pseudodelicatissima*, *Sceletonema costatum*, *Chaetoceros* sp., *Cerataulina pelagica*, *Thalassiosira* sp., *Leptocylindrus danicus*, and *Rhizosolenia alata*; (b) dinoflagellates: *Prorocentrum triestinum*, *P. minimum*, *P. micans*, *P. gracile*, *Heterocapsa* sp., and *Ceratium furca*; and (c) coccolithophorids: *Calyptrosphaera oblonga*, *Emiliania huxleyi*, and *Syracosphaera pulchra* [1]. Prior to the collection of macroaggregates, we noted the presence of marine snow on 14 June 2004 and surface foam on 15 June 2004, followed by more stable foams on 22 June 2004, as well as the presence of stringers and small aggregates on 23 June 2004, with the most intense aggregation occurring on 26 June 2004. Finally, according to observations in this study in the beginning of July, only sporadic macroaggregates were present at the sea surface.

### 2.2. Field Sampling

The macroaggregate samples were collected in the southern part of the Gulf of Trieste (45°31.46′ N; 13°33.72′ E; northern Adriatic Sea, Slovenia) during massive mucilage events in 2000 and 2004. The first sample was collected at the sea surface (0–0.5 m) on 9 June 2000 and the second sample was collected by SCUBA divers from the water column (about 12 m depth) on 1 July 2004. Both mucilage samples occurred as macrogel, either in the form of a gelatinous surface layer or as a massive, several-meter-sized, continuous cloud-like body in the water column. Their gelatinous nature allowed collecting them by hand using largemouth polyethylene bottles, taking a minimal amount (a few volumetric %) of surrounding water. The samples were immediately frozen at −80 °C pending freeze-drying. Because we used a large volume (typically 1 to 2 L) of mucilage collected in the seawater, this represents a large bulked sample, which is equivalent to multiple subsamples taken in various places and, as such, it is sufficiently representative of mucilage events in the Adriatic Sea.

The sample from 2000 was freeze-dried and subsequently washed several times with a small volume of Milli-Q water to remove sea salts. In the case of macroaggregate from 2004, the thawed mucilage was separated from the water by centrifugation at 10,000 rpm for 15 min at an ambient (25 °C) temperature. Supernatant (interstitial/pore water) and sediment (macroaggregate matrix) fractions were separated and freeze-dried. Three independent sub-samples from each freeze-dried batch were processed for multi-elemental analyses.

### 2.3. Sample Preparation and Leaching

To measure the elemental concentrations, all samples were digested in Savillex^®^ Teflon vials (Savillex, Eden Prairie, MN, USA) within individual polycarbonate compartments (A 100) containing Teflon hot plates in a clean room (class A 10,000). About 100 mg of freeze-dried material was first reacted with hydrogen peroxide (H_2_O_2_, Merck, Germany) for 24 h at an ambient temperature and further digested in 1:1 HNO_3_ (Merck, France) and HF (Merck, Germany) mixture, for 36 h at 80 °C, then in HCl (Merck, Germany) for 36 h at 80 °C, and finally treated with 1:1 HCl and HNO_3_ mixture for 36 h at 80 °C. Note that the HF addition step was omitted for the analysis of total Si. Typically, twelve Teflon reactors were processed on the hot plate. A series of reactors was composed of nine samples of macroaggregates, one certified lichen standard CRM 482 sample (from BCR, Belgium) or other NIST standards, and one blank sample. After cooling, the samples were evaporated at 70 °C for 24 h. The dry residue was dissolved in 10 mL of 10% HNO_3_ and further diluted by a factor of 10 using 2% HNO_3_ prior to analyses.

### 2.4. Elemental Analyses

The major and trace element concentrations were measured by ICP-MS (Agilent 7500 ce, Santa Clara, CA 95051, USA) using a three-point calibration against a standard solution of known concentration. Three standard solutions (1, 10, and 100 µg L^−1^ of each element in 2% HNO_3_) were measured every 10 samples. Indium and rhenium were used as internal standards to correct for instrumental drift and eventual matrix effects. The appropriate corrections for oxide and hydroxide isobaric interferences were applied for the rare earth elements (REEs). In addition to BCR CRM-482 lichen, the international geostandards of basaltic rock BE-N (from CRPG, Nancy, France), lake sediments (LKSD-3, from BCR, Belgium), Apple Leaves SRM 1515 (from NIST, Gaithersburg, MD, USA), and Pine Needles SRM 1575a (from NIST, Gaithersburg, MD, USA) were run to check the efficiency of the acid digestion protocol and the analysis. The data tables present the results of the elements, exhibiting a good agreement (±10%) between the certified or recommended values and our measurements, or for cases in which we obtain a good reproducibility (with the relative standard deviation of our various measurements of standards being lower than 10%), even if no certified or recommended data are available. During ICP MS analysis, the SLRS-5 international standard was measured at the beginning and the end of analytical session to assess the external accuracy and sensitivity of the instrument. All certified major (Ca, Mg, K, Na, Si) and trace elements (Al, As, B, Ba, Co, Cr, Cu, Fe, Ga, Li, Mn, Mo, Ni, Pb), all naturally occurring REEs (La, Ce, Pr, Nd, Sm, Eu, Gd, Dy, Ho, Er, Tm, Yb, Lu), Sb, Sr, Th, Ti, U, V, Zn), concentrations of the SLRS-5 standard, and the measured concentrations agreed with an uncertainty of 10–20%. The agreement for Cd, Cs, and Hf was between 30 and 50%. For all major and most trace elements, the concentrations in the blanks were below the analytical detection limits (≤0.1–1 ng/L for Cd, Ba, Y, Zr, Nb, REE, Hf, Pb, Th, U; 1 ng/L for Ga, Ge, Rb, Sr, Sb; and ≤10 ng/L for Ti, V, Cr, Mn, Fe, Co, Ni, Cu, Zn, As). Some rare elements, such as Sn, Nb, W, Tl, Ta, and Bi, which were not certified in the reference materials, were also measured, but their concentrations were presented only in the case when three independent subsamples provided a <20% agreement. More details about the entire analytical procedure of organic-rich soils and plants are available in Viers et al. [24,25].

The total Hg concentration in the freeze-dried powder was determined using a direct mercury analyzer (DMA-80—Milestone, Sorisole, Italy). The total mercury concentration was measured by combustion of the sample at 750 °C, pre-concentration of the Hg^0^ vapor released on a gold trap, and detection by atomic absorption. Analysis of reference materials BCR-482 (lichen, 480 ng g^−1^) and MESS-3 (sediment, 91 ng g^−1^) showed good recoveries of 450 ± 23 ng g^−1^ (*n* = 6, 1σ) and 89 ± 6 ng g^−1^ (*n* = 7, 1σ), respectively. The average uncertainty on duplicate sample analysis did not exceed 5% (1σ). Other measured components included total C and N concentrations, using catalytic combustion with Cu-O at 900 °C with an uncertainty of ≤0.5% using a Thermo Flash 2000 CN Analyzer (ThermoFisher Scientific, Hennigsdorf, Germany).

### 2.5. Data Treatment

Data of major trace element concentrations were checked for normal distribution using the Shapiro–Wilk test. Nonparametric statistical methods were used for statistical treatment. Mean and standard deviation values (mean ± standard deviation) were used to describe the uncertainty of the data. A pairwise comparative analysis was performed using the non-parametric Mann–Whitney test (U-test) to detect statistically significant differences between two independent datasets based on one given parameter (major and trace element concentration, or the elemental ratios). All graphs and figures were built using MS Excel 2016 and the Statistica-12 software package (StatSoftFrance, Maisons-Alfort, France, http://www.statsoft.com, accessed on 1 Janaury2023).

## 3. Results and Discussion

The concentrations of 57 analyzed elements are summarized in Table 1. As expected, both mucilage samples were strongly enriched in most elements relative to the aqueous fraction (except for Na, Mg, Ca, and K).

### 3.1. Interstitial Water

The concentration sequences of elements present in the mucous interstitial water are listed in Table 2. These orders indicate the degree of element accumulation by mucous matrix and relative mobility of major and trace elements. It likely stems from a combination of different biogenic and lithogenic sources and various photochemical and microbial processes occurring in the mucilage [4]. The first four most abundant elements (e.g., Na, Mg, K, and Ca) are the typical major elements of seawater. The average concentration of P_tot_ of seawater measured during summer 2004 was a concentration of about 1000× lower than that of mucous interstitial water. Del Negro et al. [26] also reported that the nutrient concentrations determined in mucous interstitial water were orders of magnitude higher than in the surrounding water. The PO_4_ concentrations are known to significantly increase in macroaggregates over time [27]. On the other hand, P can be accumulated within the aggregates over their lifetime, given that the total P concentrations of northern Adriatic sediments are rather high (0.03–0.05%), with the Mn and Fe oxides being the most significant host phases for P [21].

In contrast to the major matrix of the analyzed water sample, the concentrations of some trace elements (Ni, Cr, Nb, Dy, Sb, B, Ge, Cd, and Bi) were under the limit of detection and the presence of REEs and Th could only be qualitatively detected in the mucous interstitial water. Overall, all element concentrations were quite low in the interstitial water sample. A complex, cross-linked structure of polysaccharide-rich mucilage probably provides multiple binding sites (in organic and inorganic phases) for various elements, which are thus removed from the aqueous phase to the mucous matrix. Note that a detailed metal speciation study in the marine macroaggregates demonstrated that trace metals are bound to large macromolecules (>30 kDa), mainly glycoprotein and aminopolysaccharides [28].

### 3.2. Elemental Composition of Gelatinous Matrix

The comparison of element content in both matrix samples revealed higher concentrations of all lithogenic elements in the surface mucous sample from 2000 compared with that of the water column sample (2004). The Mann–Whitney test (U-test) demonstrated significant (*p* < 0.05) differences for all major and trace elements, except for Mg, Na, K, Sr, Se, and B. The major reason for these results is probably the higher intensity of the mucilage event of 2000 and the surface sampling location at the phase boundary of the air–sea interface, which is strongly influenced by atmospheric deposition. Overall, the mucous chemical composition is largely affected by the composition and activity of plankton biota. Ho et al. [29] reported that the cellular concentration of most tested marine eucariotic phytoplankton species (in culture) followed the order Fe > Mn > Zn > Cu > Co~Cd > Mo. A similar elemental order was observed in mucous matrices studied in this work (Table 3). The concentrations of some toxicants and micronutrients (As, Cd, Cu, Hg, Se, and Zn) were comparable to those measured in 22 plankton samples in the Adriatic [30]. Besides the taxonomic composition of phytoplankton, various environmental conditions including solar irradiance, nutrient concentration, and stoiehiometric quotas (e.g., [29]) influence the elemental stoichiometry of phytoplankton-derived particulate organic material [31]. Thus, it has long been known that, while trace metals’ biogeochemical cycles could be directly or indirectly driven by the phytoplankton metabolism [32,33], different taxonomic groups from a comparable environment could be characterized by different elemental compositions/stoichiometry [29,34,35].

In addition to metabolically controlled transport of trace metals across the microbial cell membranes, resulting in intracellular accumulation, a passive uptake based on physical adsorption, ion exchange, and chemical sorption may occur [36]. The trace metal binding sites are mostly related to the functional groups of organic constituents. Cell walls, mainly composed of polysaccharides, proteins, and lipids, comprise abundant metal binding groups such as carboxyl, sulphate, phosphate, and amino groups [36]. For example, a strong affinity for trace elements observed for diatoms, which are the main microorganisms responsible for mucilage events, is probably linked to the diatom organic coating that determines cell amphoteric characteristics and the affinity to cations and anions [37,38]. Over the residence time of mucilage in the water column, the maturation and degradation of the complex cross-linked structure of macroaggregates influences the metal uptake and binding. Thus, in the case of diatom cells, the degradation of the organic membrane could lead to a release of trapped metals into the aqueous solution [39,40].

Another important vector of trace metal control in the mucilage is inorganic phases, which are capable of accommodating a sizable amount of TE. In the northern Adriatic, various clay minerals and Fe-, P-, and Mn-rich organo-mineral phases have been indicated to be among the dominant hosts of heavy metals [21]. In addition to clays, other inorganic constituents of macroaggregates such as Fe and Mn oxides and Ca carbonates can be significant carriers of trace elements compatible with a CaCO_3_ structure (Mn, Sr, Ba, and Cd) [41]. A first-order comparison between metal content in macroaggregates and particulate matter sampled in the Gulf of Trieste [1] showed a higher content in particulate matter and surficial sediment [30], which is probably due to various efficient TE carriers present within the particulate matter [1].

### 3.3. Normalization of Macroaggregate Multi-Elemental Composition to Possible Biogenic and Lithogenic Sources

To obtain further insights into the origin of various major and trace elements in the macroaggregates, we normalized the element concentration in the mucilage samples to element concentration in the upper continental crust (UCC, [42]), mean world river suspended matter (RSM, [43]), mean total oceanic plankton (based on >120 original reports in various parts of the world ocean, [44,45]), and mean oceanic particulate suspended mater [44,45]. We consider the oceanic plankton and particulate suspended matter of the water column as the main sources of elements in the mucilage. Here, it is important to note that the phytoplankton contribution to the mucilage composition can be affected by the plankton composition itself. This impact has to be considered during normalization of the elemental composition of the mucilage to that of the plankton. Unfortunately, a full list of major and trace elements in the Adriatic plankton is not available, and there are only limited data on a few trace metals [41]. These concentrations are in reasonable agreement with the world average values as compiled by Savenko [44].

The river suspended matter reflects an essentially inorganic (mineral) source to the marine coastal zone, and even though 90% of the RSM influx is retained in the mixing zone (estuary) and only 10% is delivered to the open (pelagic) water, the RSM may represent a sizable source of major and trace elements in the coastal zones of the Adriatic Sea. In view of the lack of information on the entire elemental composition of suspended matter of local rivers of the Adriatic basin, we used the mean world river values. Finally, the UCC is used as a lithogenic source of elements, given that its average composition is similar to that of both marine and continental sediments [42].

The reasons for using average earth crust, oceanic plankton, and river suspended matter for normalizations of elemental composition of macroaggregates are multiple. First, there are no data for the full list of major and trace elements measured in this study for macroaggregates and that in Adriatic sediments and, especially, in the suspended matter of Adriatic rivers. Second, the spatial variations in average chemical composition (except several local pollutants) of lithogenic material are not high and rarely exceed 30%, which is largely sufficient for assessing the relative role of biogenic vs. lithogenic components, as stated in the Introduction of our manuscript. For example, Acquavita et al. [46] and Covelli et al. [47]) reported the following concentrations in the Adriatic Sea sediments (mean ± s.d., ppm): Li (38 ± 15), Al (4560 ± 1370), V (92 ± 29), Ni (79 ± 47), and Cr (112 ± 41). These values are fairly similar to mean river and oceanic suspended matter [44,45]: Li (30 and 20), Al (6000), V (130 and 50), Ni (84 and 70), and Cr (130 and 100). Acquiavita et al. [46] also demonstrated a relatively low (i.e., a factor of 1.0 ± 0.5) enrichment factor of Zn, Pb, Cu, Cr, and Ni concentration in the surface sediments of the Adriatic Sea compared with the regional background. Some available data on trace metal concentration in suspended particulate matter collected in the vicinity of mucuous aggregates, in the coastal zone [1,48,49], demonstrated an agreement, within the uncertainties of natural variations, of Ba, Cd, Cr, Cu, Cs, Co, Mn, Ni, and Pb with concentrations recommended by Savenko [45] for mean oceanic suspended matter. Third, spatial and temporal variations in major and trace elements river suspended matter are typically low (compared, for example, with dissolved riverine load), as demonstrated by numerous works [43,50,51,52], which allows for approximating riverine lithogenic input by one generic value.

The water column macroaggregate collected in July 2004 exhibited three groups of elements (Figure 1A): (1) strongly (>×10) impoverished in Al, Si, Sc, Ti, V, Cr, Mn, Fe, Co, Ni, Cu, Ga, Ge, Rb, Y, Zr, Nb, Cs, Ba, REEs, Hf, Ta, W, Th, and U compared with the earth crust or river suspended material; (2) anomalously enriched in B, Na, Ca, and Se relative to the UCC; and (3) elements whose concentrations were similar to those of the RSM (Li, Mg, P, Zn, As Sr, Mo, Cd, Sn, Sb, and Bi). Normalization of elemental composition of the water column macroaggregate to that of oceanic plankton and suspended particles demonstrated that these two sources are much more similar to the studied mucous material (Figure 1B). In fact, most elements in the mucilage are within the same order of magnitude as mean oceanic plankton. At the same time, the mucilage is strongly (×10–20) enriched in B, Al, Sc, and Mn and impoverished in Si, P, Cs, and Ba relative to the plankton, as confirmed by U-test at *p* < 0.05. Some ‘anomalously’ enriched elements (Al and Mn) can be better approximated by considering the combination of oceanic plankton and marine suspended particulate matter. The lithogenic elements in the water column macroaggregate were much closer to the plankton rather than to inorganic sources such as UCC or RSM (Figure 2). The crust and river suspended matter normalized REE patterns were flat, with some weak mid REE (Eu-Gd) maximum, probably linked to the presence of a feldspar source. Note that a local Sm-Eu maximum of the plankton-normalized pattern could be an artefact owing to the high uncertainty of these element concentrations in mean oceanic plankton (i.e., [44]).

The surface macroaggregate sampled in June 2000 was more enriched in lithogenic component compared with the water column sample. The majority of elements (e.g., Li, Mg, Al, K, Cr, Mn, Fe, Co, Ni, Cu, Zn, Ga, Ge, Se, Rb, Zr, Nb, Mo, Cd, Sb, REEs, Hf, Ta, W, Pb, Bi, Th, and U) were located between the UCC- or RSM-normalized pattern (Figure 3A) and plankton- or particle-normalized pattern (Figure 3B), although it can be seen that mean oceanic suspended particulate matter could explain the concentration of the largest number of major and trace elements, which included both labile, highly soluble cations (Li, Mg, K, Sr, and Ba) and anions (Mo, Cr, and Se), divalent transition metals (Mn, Co, Ni, and Zn), V, hydrolysates (Al, Fe, Ga, Y, Sn, and Hf), Bi, and U. The REE patterns of surface macroaggregates were generally flat (Figure 4). The sample was strongly enriched in REE relative to oceanic plankton, and both the shape and magnitude of REE pattern could stem from a roughly 1:1 combination of oceanic suspended particles and river suspended matter.

Overall, a tight association between organic matter of plankton exometabolites and mineral particles, as demonstrated by trace element geochemical analyses in this study, corroborates the importance of such associations in the mucilage phenomenon (including its formation, evolution, and transformation), as well as spatial and seasonal distribution and the accumulation of macroaggregates [53].

A limitation of the present study is the rather small number of analyzed mucilage samples. However, based on numerous measurements of our group on mucilage aggregates of the Adriatic Sea using other techniques [1,2,3,4,13,14,15,16,20,21], we are confident that (1) the collected samples are representative of the mucilage event of the specific month and year, and (2) spatial variations in other parameters of mucilage composition are not highly pronounced, provided that we clearly distinguish the surface and water column aggregates. Indeed, it is known that the horizontal and vertical distribution and accumulation of macroaggregates is heterogeneous and time-dependent on different factors such as the conditions in the water column; the size, form, and composition of the aggregates; and the environmental conditions. However, the same (typical) characteristic biological and chemical compositions of northern Adriatic macroaggregates were observed during all such events that have been largely described and confirmed by various authors [1,9,12,14,18,27,53]. More specifically, the variations in the chemical composition of macro-aggregates are rather low, with the standard deviation rarely exceeding a few percent of the average, in terms of macro composition. This is illustrated by a systematic study of the mean organic composition of macroaggregates collected over several seasons in the Pesaro region of the Adriatic Sea [14,53]. These authors demonstrated very low (i.e., ≤0.7%) inter-annual variations in the aggregate composition. Further, they also demonstrated a rather stable chemical composition of both seawater and macroaggregates sampled in the Gulf of Trieste over the spring–summer–autumn seasons in 2004, with typical variations not exceeding ±30%. Therefore, we believe that the selected samples are representative of the mucilage event of a specific month and year and, in general, for Northern Adriatic macroaggregates. At the same time, we have to admit that detailed, spatially resolved sampling of mucilage events across seasons for multi-elemental analyses is certainly necessary, although it represents sizable, if not cost-prohibitive, research efforts that extend beyond the scope of the present study.

## 4. Conclusions

The results of this study demonstrate that a massive mucilage event produces mucuous material that is strongly enriched in a number of macro and micronutrients, trace elements, and toxicants, similar to that of marine plankton and oceanic suspended particles, with an additional input of lithogenic mineral material. Therefore, a huge amount of macroaggregates produced during these events contribute to trace element removal, accumulation, and sedimentation and, consequently, to their cycling and immobilization in the northern Adriatic basin. We believe that it is the dual nature of the macroaggregates, i.e., heterogeneous cross-linked polysaccharide-rich organic matrix combined with mineral (inorganic) constituents, that mostly contributes to element transformation, behavior, and cycling in the northern Adriatic.

Analyzing 57 major and trace element allowed the global assessment of the chemical signature of the mucous material and helped to reveal its origin and possible transformation pathways. To the best of our knowledge, there is no similar work on other exometabolites of living organisms. In this study, we took the advantage of a really massive mucilage event that could provide sufficient material for both extraction and preparation (centrifugation, leaching, and freeze-drying) and analyses. The main implication of the analysis is resolving two major contributions to marine organisms metabolites: the lithogenic (“external”) source, reflected by the abundance and pattern of biologically inert geochemical tracers, such as, for example, rare earth elements, and the “autochthonous” source of micronutrients, exerted by live cells together with organic polymers or taken up via the absorption of ions from the surrounding seawater.

The element contents in both the solid and aqueous fraction of the macroaggregates reflected the binding capacity of mucilage matrix and the binding strength/exchange capacity of trace metals. As hypothesized, the elemental composition was different between surface and water column macroaggregates. Furthermore, we also confirmed that the trace element composition of macroaggregates, including the rare earth element pattern, reflected a combination of terrestrial (allochthonous) sources of silicate material, marine suspended particles, and micronutrients of biogenic origin. In this regard, the mean elemental composition of selected samples was related to the biological and chemical components of the macroaggregates and stem from a combination of autochthonous, dominantly biotic (plankton exometabolites), and allochthonous (mineral and organic riverine, atmospheric, and anthropogenic) inputs occurring during the mucilage events. It is possible that the composition and amount of allochthonous input are controlled by the duration of the mucilage event, meteorological conditions, and hydrodynamics of the water column; however, our study could not quantify the impact of these factors.

Further, the enrichment of various elements in the interstitial water of the mucilage clouds (aqueous supernatant) relative to the surrounding seawater could be linked to photochemical and microbial processes occurring within the macroaggregates. At the same time, mucous scavenging and accumulation of various particles and chemicals from seawater contribute to a decrease in the concentration of these substances in ambient seawater, owing to the high adsorbing capacity and abundant binding sites present within the macroaggregates.

The main limitation of this work is the rather small number of samples, despite their sufficient representability. In order to quantify the source of lithogenic (i.e., mineral-related Ca, Si, Sr, and Fe) or biogenic (micronutrients such as Zn, Cu, Ni, Se, and Mo) and external pollutants (Cd, Pb, Hg) in the mucilage elemental signatures, high-resolution stable isotope measurements using multi-collector ICP MS are necessary. In addition, better spatial and seasonal coverage of sampling, currently cost-prohibitive for multi-elemental composition of macroaggregates, should provide further insights into the short-term and small-scale variability in marine microorganism exometabolites.

## Figures and Tables

**Figure 1 metabolites-13-00726-f001:**
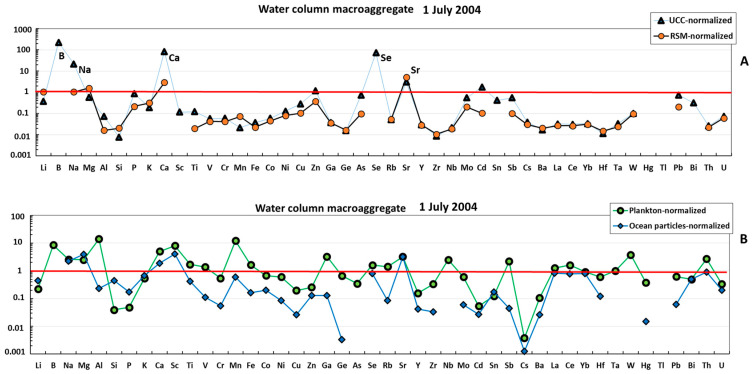
UCC- and RSM-normalized (**A**) and plankton- and ocean-particle-normalized (**B**) elemental abundance in the mucilage from the water column. The UCC, RSM, oceanic plankton, and oceanic particles data are from [42,43,44,45], respectively.

**Figure 2 metabolites-13-00726-f002:**
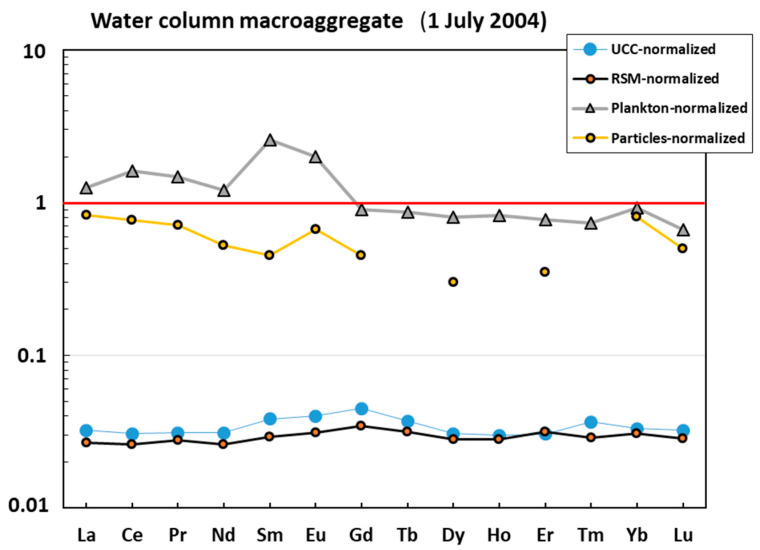
UCC-, RSM-, and plankton- and ocean-particle-normalized [42,43,44,45] rare earth element (REE) pattern in the mucilage from the water column.

**Figure 3 metabolites-13-00726-f003:**
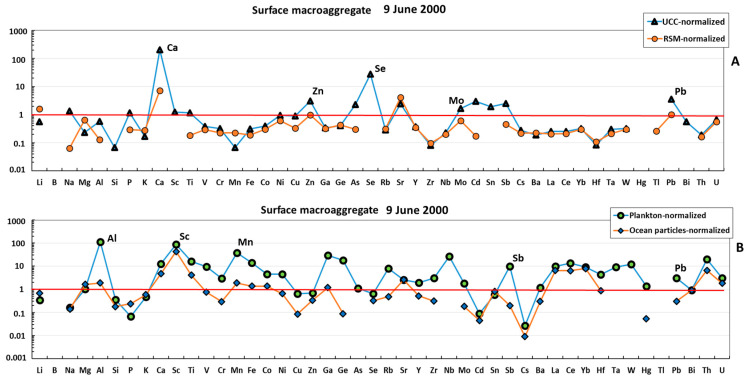
UCC- and RSM-normalized (**A**) and plankton- and ocean-particle-normalized (**B**) elemental abundance in the mucilage from the surface layer. The UCC, RSM, oceanic plankton, and oceanic particle data are from [42,43,44,45], respectively.

**Figure 4 metabolites-13-00726-f004:**
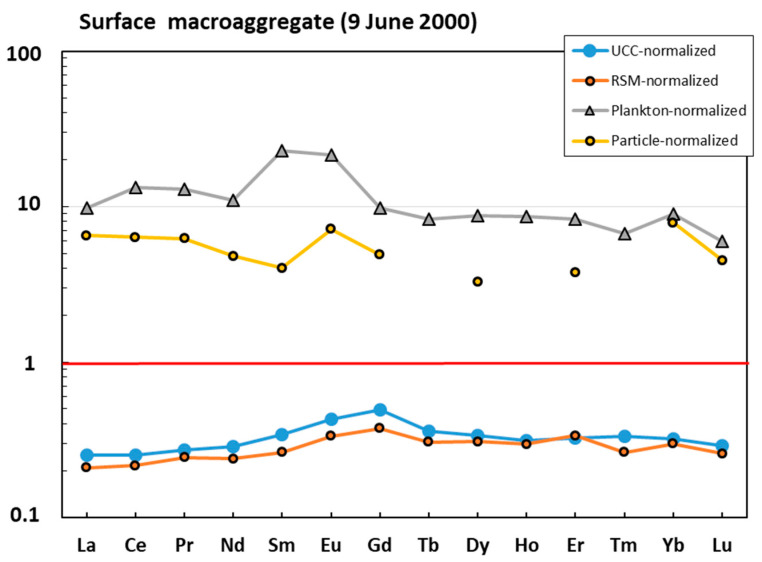
UCC-, RSM-, and plankton- and ocean-particle-normalized [42,43,44,45] rare earth element (REE) pattern in the mucilage from the surface layer.

**Table 1 metabolites-13-00726-t001:** Elemental composition of macroaggregates (three independent subsamples of each mucilage event); mean ± standard deviation, µg g^−1^.

Element	Macroaggregate—Surface 9 June 2000	Macroaggregate—Water Column 1 July 2004, Aqueous Fraction	Macroagregate—Water Column 1 July 2004
Li	13.5 ± 1.5	5.14 ± 0.14	8.96 ± 0.33
B	<d.l.	<d.l.	429 ± 75
Na	5600 ± 58	206,000 ± 15,000	90,500 ± 970
Mg	8070 ± 120	26,500 ± 640	19,900 ± 290
Al	11,100 ± 150	14 ± 1	1400 ± 40
Si	20,800 ± 470	276 ± 20	2350 ± 96
P	590 ± 17	9 ± 2	436 ± 14
K	4620 ± 160	13,280 ± 535	5330 ± 150
Ca	186,000 ± 5000	12,200 ± 450	74,900 ± 3000
Sc	17.2 ± 0.46	0.43 ± 0.04	1.62 ± 0.09
Ti	811 ± 29	1.05 ± 0.2	85 ± 3
V	37.5 ± 0.68	0.41 ± 0.02	5.50 ± 0.10
Cr	29 ± 0.84	<d.l.	5.42 ± 0.14
Mn	377 ± 17	5.75 ± 0.28	121 ± 5
Fe	10,900 ± 340	79 ± 3	1300 ± 45
Ni	45 ± 2.8	<d.l.	6.1 ± 0.8
Co	6.8 ± 0.09	0.058 ± 0.005	1.0 ± 0.034
Cu	25.3 ± 0.52	2.75 ± 0.10	7.8 ± 0.2
Zn	203 ± 3.5	108 ± 3	77.3 ± 0.93
Ga	5.86 ± 0.12	0.09 ± 0.01	0.65 ± 0.05
Ge	0.53 ± 0.02	<d.l.	0.02 ± 0.01
As	11.0 ± 0.28	0.41 ± 0.04	3.49 ± 0.14
Se	2.5 ± 0.4	6.08 ± 0.55	6.47 ± 0.30
Rb	24.0 ± 0.4	3.27 ± 0.12	4.23 ± 0.11
Sr	770 ± 15	185 ± 6	964 ± 17
Y	7.75 ± 0.13	0.0094 ± 0.002	0.625 ± 0.013
Zr	15.3 ± 0.3	0.02 ± 0.005	1.67 ± 0.07
Nb	2.7 ± 0.09	<d.l.	0.25 ± 0.02
Mo	1.8 ± 0.2	0.3 ± 0.05	0.6 ± 0.05
Cd	0.27 ± 0.03	<d.l.	0.16 ± 0.06
Sn	4.1 ± 0.2	0.13 ± 0.02	0.87 ± 0.13
Sb	0.99 ± 0.03	<d.l.	0.22 ± 0.03
Cs	1.35 ± 0.03	0.01 ± 0.003	0.19 ± 0.02
Ba	118 ± 10	1.64 ± 0.05	10.6 ± 0.13
La	7.79 ± 0.10	0.01 ± 0.002	1.00 ± 0.03
Ce	15.8 ± 0.35	0.02 ± 0.001	1.93 ± 0.04
Pr	1.93 ± 0.05	<d.l.	0.22 ± 0.02
Nd	7.7 ± 0.15	0.013 ± 0.002	0.84 ± 0.03
Sm	1.6 ± 0.05	0.01 ± 0.003	0.18 ± 0.02
Eu	0.43 ± 0.05	<d.l.	0.04 ± 0.005
Gd	1.96 ± 0.05	0.01 ± 0.002	0.18 ± 0.015
Tb	0.25 ± 0.02	0.0015 ± 0.0004	0.023 ± 0.005
Dy	1.31 ± 0.04	<d.l.	0.12 ± 0.02
Ho	0.26 ± 0.08	<d.l.	0.03 ± 0.005
Er	0.75 ± 0.05	<d.l.	0.07 ± 0.015
Tm	0.10 ± 0.02	<d.l.	0.01 ± 0.002
Yb	0.63 ± 0.02	<d.l.	0.05 ± 0.005
Lu	0.09 ± 0.01	<d.l.	0.01 ± 0.003
Hf	0.44 ± 0.02	0.01 ± 0.002	0.06 ± 0.01
Ta	0.27 ± 0.08	<d.l.	0.03 ± 0.005
W	0.60 ± 0.05	0.01 ± 0.003	0.19 ± 0.03
Tl	0.14 ± 0.03	<d.l.	<d.l.
Pb	61 ± 1	0.92 ± 0.06	12.4 ± 0.8
Bi	0.09 ± 0.02	<d.l.	0.05 ± 0.015
Th	1.98 ± 0.04	<d.l.	0.27 ± 0.02
U	1.82 ± 0.03	0.06 ± 0.002	0.20 ± 0.015
Hg	0.27 ± 0.02	n.d.	0.075 ± 0.008

**Table 2 metabolites-13-00726-t002:** Order of element abundance in the mucous interstitial water.

Concentration Range	Elemental Order/Sequence
>10,000 ppm	Na > Mg > K > Ca
>10 ppm	Si > Sr > Zn > Fe > Al
>1 ppm	P > Se > Mn > Li > Rb > Cu > Ba > Ti
<1 ppm	Pb > Sc, V, As > Mo > Sn > Ga > Co, U

**Table 3 metabolites-13-00726-t003:** Comparison of the order of element abundance in the mucuos macroaggragates collected at the surface (June 2000) and at a depth of 12 m within the water column (July 2004). The differences are significant for all indicated elements (*p* < 0.05, U-test), except for Mo, Th, and U.

Sampling Site	Date	Elemental Order/Sequence
Surface	June 2000	Al > Fe > Ti > Mn > Zn > Ba > Pb > Ni > V > Cr, Cu > Co > U, Mo, Th > Cd
Water column	July 2004	Al > Fe > Mn > Ti > Zn > Pb, Ba > Cu > Ni, V, Cr > Co > Mo, U, Th, Cd

## Data Availability

All of the obtained data are contained within the article (Table 1).

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
