# Peer review of "Elemental Composition of Plankton Exometabolites (Mucous Macroaggregates): Control by Biogenic and Lithogenic Components"

_metabolites, 2023, doi:10.3390/metabo13060726_

Round 1

Reviewer 1 Report

The title of the presented review corresponds to the content of the article. The authors of this article present an interesting study on the elemental composition of mucous macroaggregates. In my opinion, the authors planned the experiment well and used a sensitive analytical technique, which is described in sufficient detail in the dedicated chapter. The text is well structured, easy to read and understand. The discussion of the results is clear and well interpreted.

Nevertheless, please clarify/correct:

1.      Collection year for the surface macroaggregate 2000 (2.2 Sampling ) or 2001 (Table1)?

2.      To facilitate understanding of Table1, please explain each subcolumn.

3.      Please correct the nr of decimals in Table 1

4.      Table 1, 3rd column misprint: macroaggregate

5.      Figure 1A misprint: UCC normalized

6.      Order of the figures

Author Response

Reviewer No 1

The title of the presented review corresponds to the content of the article. The authors of this article present an interesting study on the elemental composition of mucous macroaggregates. In my opinion, the authors planned the experiment well and used a sensitive analytical technique, which is described in sufficient detail in the dedicated chapter. The text is well structured, easy to read and understand. The discussion of the results is clear and well interpreted.

We appreciate positive evaluation of our work and revised the text, figures and tables as indicated by the reviewer.

Nevertheless, please clarify/correct:

  1. Collection year for the surface macroaggregate 2000 (2.2 Sampling ) or 2001 (Table1)?

This is 2000 as stated in the text; corrected in the table accordingly.

  1. To facilitate understanding of Table1, please explain each subcolumn.

We re-organized and compacted this table, presenting the mean value ± standard deviation.

  1. Please correct the nr of decimals in Table 1

We carefully edited the numbers, verified the mean values and removed unjustified decimals.

  1. Table 1, 3rd column misprint: macroaggregate

Fixed

  1. Figure 1A misprint: UCC normalized

Corrected; thanks for catching this!

  1. Order of the figures

We reorganized the presentation as necessary 

Reviewer 2 Report

The authors analyzed the composition of mucous macroaggregates in a few samples collected in the Adriatic Sea. Based on these analyses, it is concluded that the composition of the macroaggregates is similar to the mean composition of the oceanic plankton. The formation of macroaggregates is an important ambient problem in the Adriatic Sea and any information about their origin along with the factors that contribute to their appearance might be relevant. Consequently, the topic is of interest. Additionally, the analytical methods are sound and the manuscript is well written. However the work has serious flaws that reduce its interest for publication:

 (1) The work is based only on three samples which is not enough to perform robust assessments based on statistical analyses. I guess that the composition of mucous macroaggregates would be variable spatially even during a single event. With the data shown, it is unclear which the representativeness of the three analyzed samples is and how the results can be extrapolated to other events.

(2) The work lacks statistical analyses that are necessary to assess the differences between the two sampled events as well as between the phytoplankton and macroaggregates composition. Without these analyses, the conclusions are speculative.

(3) The composition of the macroaggregates is not compared with the composition of the phytoplankton and lithogenic mineral material in the sampling area. The authors performed these comparisons based on global data that may not match the features of the plankton in the study area. In fact, the authors highlighted in the Introduction that the macroaggregates have a heterogeneous composition that is influenced by different local sources.

(4) The main conclusion (i.e. the composition of the macroaggregates indicates that they are produced mainly by the phytoplankton) is not novel. The work reinforces findings that are well established and it is unclear which the new addressed point is.  

Other specific comments:

(5) Lines 91-94. The work would become more attractive if some specific hypothesis to be tested is formulated.

(6)   Lines 104-114. These comments about the sampled events should be supported with references.

(7) Lines 116-124. The exact location of the sampling stations should be indicated.

(8) Line 185. Some deviation measurement (e.g. standard deviation) should be shown in the table.

(9) Lines 214-219. These differences must be tested with some statistical analysis.

(10) Line 225. I am not sure that a reference from 1978 is relevant to compare with the results obtained that are based on samples collected in 2000 and 2004.

(11) Line 270. I doubt that “the mean world river values” are appropriate to perform these comparisons. The features of the rivers in the study area would differ significantly from these mean values.

Author Response

We thank the reviewer for overall positive evaluation of our research efforts and we carefully revised the text and presented our arguments to the reviewer.

See attached pdf file

Reviewer 3 Report

The manuscript entitled : "Elemental composition of plankton exometabolites (mucous macroaggregates): control by biogenic and lithogenic compo nents" shows a study that is certainly interesting but needs major revision in the structure of the text (some grammatical errors, periods too long and sentences sometimes difficult to read).Moreover, there are a few points to be adressed:

-Please add the sampling methods 

-Line 145: What is the composition of the standard solution?

-Lines 154-155: Please rewrite this sentence,  it's not clear

-Table 1: 2s n=3 , this value have no reference, what it does mean?

-Figure Legend have to be reworked, in Figure (1-2-3-4) the values lack of a data reference about the measure.

-Figure 4 is figure 3 and viceversa

-Line 322: What is the "aquatorium" which the authors are referring to?

Author Response

Reviewer No 3

The manuscript entitled : "Elemental composition of plankton exometabolites (mucous macroaggregates): control by biogenic and lithogenic components" shows a study that is certainly interesting but needs major revision in the structure of the text (some grammatical errors, periods too long and sentences sometimes difficult to read).

We thank the reviewer for positive evaluation of our work. We carefully edited the text, corrected grammatical issues, revised, clarified and shortened a number of sentences.

Moreover, there are a few points to be adressed:

-Please add the sampling methods 

As stated in the text, the mucilage samples occurred as macrogel, either in the form of gelatinous surface layer or as massive, several meter-sized, continuous cloud-like body in the water column. Their gelatinous nature allowed collecting them by hand using polyethylene bottles, taking minimal amount (a few volumetric %) of surrounding water.

-Line 145: What is the composition of the standard solution?

Three standard solutions (1, 10 and 100 µg L-1 of each element in 2% HNO3) were measured every 10 samples. Added to the text accordingly.

-Lines 154-155: Please rewrite this sentence,  it's not clear

Shortened and revised for clarity. The 10 % agreement was accepted as good recovery.

-Table 1: 2s n=3 , this value have no reference, what it does mean?

Here, 2s is standard deviation; now added as ± to the mean value and explained in the Table caption.

-Figure Legend have to be reworked, in Figure (1-2-3-4) the values lack of a data reference about the measure.

We revised all figures as necessary and added relevant references in the figure legends.

-Figure 4 is figure 3 and viceversa

We replaced the position of figures in the text, as indicated by this and other reviewers.

-Line 322: What is the "aquatorium" which the authors are referring to?

Adriatic basin; revised.

Reviewer 4 Report

Kovac et al. used ICP-MS to investigate the multi-elemental composition of northern Adriatic macroaggregates, comparing it to average oceanic plankton and lithogenic sedimentary material, and assessing the differences in the element content of their water (interstitial) and gelatinous (matrix) fractions. The author compares data from 2000 and 2004. Overall, the research is both fascinating and overwhelming. However, certain improvements must be made to improve the manuscript.

1.    Please explain why the author feels the necessity to investigate the multi-element composition of northern Adriatic macroaggregates over a certain time period. Why is it so crucial? Please include more information in the introduction.

2.    Please provide more information about the location of the samples. (Specific coordinates). It would be fantastic if the author provided a map/figure of the northern Adriatic Sea, where the samples were collected. Is the sample taken in 2000 and 2004 at the same coordinate?

3.    The method part is overly broad; if possible, the author should divide it into more specialized sections.

4.    Did the author collect secondary data as well? Temperature, pH, light exposure, organisms in the vicinity of the sampling site, and so on. If so, the author recommends include it in the manuscript.

5.    Why are the sampling sites different in 2000 and 2004 (surface vs. water column)? Please elaborate.

6.    The author did not cite all the figures in the manuscript's body.

7.    Because the topic is about metabolites, it would be great if the author could include data on metabolites using HR-ESI-MS or GC-MS. If the author is unable to give them, please explain why and why the data is enough to present in the Metabolites.

8.    Please revise the conclusion, considering the author's goal and the outcome.

9.    Please also include available supplementary data, including chromatograms.

10. Please also include the limitations of the work.

Author Response

Reviewer No 4

Kovac et al. used ICP-MS to investigate the multi-elemental composition of northern Adriatic macroaggregates, comparing it to average oceanic plankton and lithogenic sedimentary material, and assessing the differences in the element content of their water (interstitial) and gelatinous (matrix) fractions. The author compares data from 2000 and 2004. Overall, the research is both fascinating and overwhelming. However, certain improvements must be made to improve the manuscript.

We appreciate positive evaluation of our work and revised the text following the recommendations of the reviewer.

  1. Please explain why the author feels the necessity to investigate the multi-element composition of northern Adriatic macroaggregates over a certain time period. Why is it so crucial? Please include more information in the introduction.

Massive production of plankton exometabolites in the Northern Adriatic does not happen each year, and it still remains an unpredictable event. For this reason, we possessed only limited amount of material for analyses. We totally agree that multi-annual, decadal scale study would be necessary but it becomes cost-prohibitive in terms of high-resolution multi-elemental analyses such as those reported in the present manuscript.

Within the context of Metabolite content and in particularly, the Special Issue, in this work, we addressed, for the first time, multi-element chemical composition of plankton exometabolites.  In the revised Introduction, we also stressed that, although rather small number of samples cannot allow quantifying the partial contribution of each source, this first study of multi-elemental composition of plankton exometabolites provides a background for further, highly spatially and seasonally resolved research. We revised the Introduction accordingly.

  1. Please provide more information about the location of the samples. (Specific coordinates). It would be fantastic if the author provided a map/figure of the northern Adriatic Sea, where the samples were collected. Is the sample taken in 2000 and 2004 at the same coordinate?

This is very valuable comment. The macroaggregate samples were collected in the southern part of the Gulf of Trieste (45°31,46’ N; 13°33,72’ E; northern Adriatic Sea, Slovenia) during massive mucilage events 2000 and 2004. Therefore, both sampling campaign were performed in approximately the same (± 100 m) location. We revised the text as requested. We believe that addition of a map is not useful because there is only one sampling site, where two mucilage events have occurred. Note that the mucilage does not appear each year.

  1. The method part is overly broad; if possible, the author should divide it into more specialized sections.

We agree and revised this section  as recommended; we also added necessary part on statistical treatment. The Method section is now composed of 5 subsections:

2.1. Mucilage events

2.2. Field sampling

2.3. Sample preparation and leaching

2.4. Elemental analyses

2.5. Data treatment

  1. Did the author collect secondary data as well? Temperature, pH, light exposure, organisms in the vicinity of the sampling site, and so on. If so, the author recommends include it in the manuscript.

Extensive description of external environment during mucilage sampling event including the seawater microorganisms are provided in ref. [1]. Northern Adriatic macroaggregates are characterized by a heterogenous composition, comprising phytoplankton, bacteria and cyanobacteria, mesozooplankton and microzooplankton and zooplankton debris (i.e. crustacean cuticles and antennae, faecal pellets), yeasts, pollen and various inorganic components such as the empty frustulaes of diatoms and skeletal remains of coccolithophorids, empty thecae of dinoflagellates and mineral particles (Kovac et al., 2008, ref. 1). Diatoms are usually reported as the dominant group of macroaggregates,  but also other groups, such as dinoflagellates, microflagellates and coccolithophorids are present. This information is provided in the 2nd paragraph of revised Introduction. Thus, the sampled 2000 and 2004 events were accompanied by development of three dominating plankton a) diatoms: Cylindrotheca closterium, Cyclotella sp., Pseudo-nitzschia pseudodelicatissima, Sceletonema costatum, Chaetoceros sp., Cerataulina pelagica, Thalassiosira sp., Leptocylindrus danicus, Rhizosolenia alata; b) dinoflagellates: Prorocentrum species such as P. triestinum, P.minimum, P. micans, P. gracile, Heterocapsa sp., Ceratium furca, Gymnodinium-like dinoflagellate; c) coccolithophorids: Calyptrosphaera oblonga, Emiliania huxleyi and Syracosphaera pulchra [ref. 1]. However, we are reluctant to provide such a detailed (and already published) information in the revised manuscript.

  1. Why are the sampling sites different in 2000 and 2004 (surface vs. water column)? Please elaborate.

The main reason for this was logistical difficulty of collecting water sample via scuba divers in 2000. However, the sampling was representative for both events.

Indeed, it is known that the horizontal and vertical distribution and accumulation of macroaggregates is heterogeneous and time-dependent on different factors such as the conditions in the water-column, the size, form and composition of the aggregates and the environmental conditions. However, the same (typical) characteristic biological and chemical composition of northern Adriatic macroaggregates were observed during every such event that have been largely described and confirmed by various authors (Stachowitsch, 1990;  Posedel in Faganeli, 1991; Herndl, 1992; Degobbis et al., 1995;  1998; Marchetti et al., 1989;  Fonda Umani  et al. 1989; Faganeli et al., 1995, 1998, 2002, 2004; Mingazzini, 1995; Giani et al., 1995; Vollenweider and Rinaldi, 1995; Baldi et al., 1997; Herndl et al., 1999; Penna et al., 1993, 2000, 2003, Manganelli  and Funari, 2003; Cozzi et al., 2004; Kovač et al., 2005; Mecozzi et al., 2005 etc.). Therefore, we believe that the selected samples, be it surface or water column, are representative for the mucillage event of specific month and year.

  1. The author did not cite all the figures in the manuscript's body.

We carefully verified the citations of the figures in the text and strongly modified their position and layout.

  1. Because the topic is about metabolites, it would be great if the author could include data on metabolites using HR-ESI-MS or GC-MS. If the author is unable to give them, please explain why and why the data is enough to present in the Metabolites.

Good point. Mucuous macroaggregates of the Gulf of Trieste of two specific mucilage events (2000 and 2004) are extensively described from the view point of organic chemistry, including solid-state 13C NMR, FT-IR, 1H NMR, Cryo-SEM (Kovac et al., 2002, 2004, 2005; Faganeli et al., 2010; Turk et al., 2010 – refs 1-4, 10, 13, 15, 16 and 20 of the revised text). Carbohydrate contents in the mucilage was assessed via 3-methyl-2-benzothiazoline hydrazine hydrochloride (MBTH), 2,4,6-tripyridyl-s-triazine (TPPZ) assays; characterization of oligosaccharides using HPLC/RI revealed maltose and pentaose as the main component (Penna et al., 2003; ref. 14 of the revised text). The high molecular weight of water-soluble fraction of mucous macroaggregates was confirmed by the size exclusion chromatography (SEC). Four major classes of structural elements of macroaggregates were identified: carbohydrates, ester and amide functional groups, aliphatic and organosilicon components. The spectroscopic analyses showed the same general structural pattern of “fresh” and more aged macroaggregates samples indicating the preservation of organic matter during mucilage event [1]. We added most important information in the revised text; however, we believe that it is not reasonable reproducing detailed former findings in the current, original, manuscript. Note that it is the background on the metabolite chemistry acquired in numerous former works that allowed us to perform a novel study of multi-elemental composition.

  1. Please revise the conclusion, considering the author's goal and the outcome.

We strongly revised the conclusions, via presenting and discussing the initial hypotheses of this study. Note that the Introduction was also revised, via adding specific goals and testable hypotheses. We also added limitations of the work in revised Conclusions, see our response to comment # 10.

  1. Please also include available supplementary data, including chromatograms.

The ICP MS analyses performed in the present study provide the count per second values for each isotope, but there is no graphical illustration of analytical output other than the calibration curves for each element, corrected for internal In-Re standard. According to the common practice in geochemistry, these technical data of the instrument are not presented. Therefore, all the necessary analytical results are listed in relevant Table 1 (which was strongly revised for clarity). As for the organic (metabolite per se) component of the studied samples, as indicated above (response to comment No 7), it is unreasonable to reproduce already published data in new manuscript.

  1. Please also include the limitations of the work.

We thank the reviewer for this valuable suggestion and added the following:  

“The main limitations of this work are rather small number of samples, despite their sufficient representability. In order to quantify the source of lithogenic (i.e., mineral-related Ca, Si, Sr, Fe) or biogenic (micronutrients such as Zn, Cu, Ni, Se, Mo) and external pollutants (Cd, Pb, Hg) in the mucilage elemental signatures, high resolution stable isotope measurements using multi-collector ICP MS are necessary. In addition, better spatial and seasonal resolution of sampling, currently cost-prohibitive for multi-elemental composi-tion of macroaggregates, should provide further insights into short-term and small-scale variability of marine microorganism exometabolites.”

Reviewer 5 Report

Elemental composition of plankton exometabolites (mucous macroaggregates): control by biogenic and lithogenic components

Manuscript Number: Metabolites-2259645

However, having thoroughly reviewed the manuscript presented to me, I have some major comments and suggestions, which I present below:

1.      The manuscript suffers from lots of grammatical and spelling mistakes.

2.      Why was the mucilage sampling done in 2000 and 2004 and not after that? The mucilage sampling depth of 2000 is missing.

3.      How many mucilage samples were collected during each sampling event?

4.      What is the reason for doing so many elemental composition of so macroaggregates? What is the implication of this analysis?

5.      I can’t find any statistics in the results.

6.      The discussion part is also very poor.

7.      The Conclusion is out of focus and not supported by results.

8.      What is the main implication of this study?

Author Response

Reviewer No 5

However, having thoroughly reviewed the manuscript presented to me, I have some major comments and suggestions, which I present below:

  1. The manuscript suffers from lots of grammatical and spelling mistakes.

We thank the reviewer for pointing this out. We have carefully edited the text for English grammar and spelling.

  1. Why was the mucilage sampling done in 2000 and 2004 and not after that? The mucilage sampling depth of 2000 is missing.

Good point. There was no mucilage after 2004. Already in the beginning of July, only sporadic macroaggregates were present at the sea surface and we considered them as not representative for massive mucilage effect. In 2000, we sampled surface macroaggregate from 0-0.5 m depth; added to revised section 2.2.

  1. How many mucilage samples were collected during each sampling event? Three individual subsamples as indicated in the text (section 2.2).

  1. What is the reason for doing so many elemental composition of so macroaggregates? What is the implication of this analysis?

The very interest of this study is a first-time assessment of total multi-elemental (50 elements, from lithium to uranium) composition of planktonic exometabolites. Analyzing all, not several selected, major and trace element allows global assessment of chemical signature of the material and helps to reveal its origin and possible transformation pathways. To the best of our knowledge, there is no similar study on other exometabolites of living organisms. Here we took the advantage of really massive mucilage event that could provide sufficient material for both extraction and preparation (centrifugation, leaching, freeze-drying) and analyses.

The main implication of analysis is resolving two main contributions to marine organisms metabolites: lithogenic (“external”) source, reflected by abundance and pattern of biologically-inert geochemical tracers, such as for example, rare earth elements, and “autochthonous” source of micronutrients, exerted by live cells together with organic polymers or taken up via absorption of ions from surrounding seawater. We added this implication in the revised text (conclusions)

  1. I can’t find any statistics in the results.

This is very pertinent remark, and we strongly revised the text and tables in response to this comment.

Data of major trace element concentrations were checked for normal distribution using the Shapiro-Wilk test. Nonparametric statistics methods were used for statistical treatment. Mean and standard deviation values (mean ± standard deviation) were used to de-scribe the uncertainty of the data. A pairwise comparative analysis was performed using the non-parametric Mann-Whitney test (U-test) to detect statistically significant differences between two independent datasets based on one given parameter (major and trace element concentration, or the elemental ratios).

In Table 1, we explained that we present ); the mean ± standard deviation.

In section 3.2, we added: “Comparison of element content in both matrix samples revealed higher concentrations of all lithogenic elements in surface mucous sample from 2000 compared to that of the water column sample (2004). The Mann-Whitney (U-test) demonstrated significant (p < 0.05) differences for all major and trace elements except Mg, Na, K, Sr, Se and B.”

In the revised Table 3, we noted that the differences are significant for all indicated elements (p < 0.05, U-test) except Mo, Th and U.

  1. The discussion part is also very poor.

In this manuscript, we chose to combine the presentation of Results and Discussion, which is compatible with format of the journal. In the revised version, however, we strongly extended the section 3.3. of the discussion and added a big deal of new references.

  1. The Conclusion is out of focus and not supported by results.

We strongly revised the Conclusions and added the following:

Analyzing 57 major and trace element allowed global assessment of chemical signature of the mucous material and helped to reveal its origin and possible transformation pathways. To the best of our knowledge, there is no similar work on other exometabolites of living organisms. In this study, we took the advantage of really massive mucilage event that could provide sufficient material for both extraction and preparation (centrifugation, leaching, freeze-drying) and analyses. Element contents in both solid and aqueous fraction of the macroaggregates reflected the binding capacity of mucilage matrix and the binding strength / exchange capacity for trace metals. As it was hypothesized, elemental composition was different between sur-face and water column macroaggregates. Furthermore, we also confirmed that trace element composition of macroaggregates, including Rare Earth Element pattern, reflected a combination of terrestrial (allochthonous) sources of silicate material, marine suspended particles, and micronutrients of biogenic origin.

  1. What is the main implication of this study?

The main implication of analysis is resolving two main contributions to marine organisms metabolites: lithogenic (“external”) source, reflected by abundance and pattern of biologically-inert geochemical tracers, such as for example, rare earth elements, and “autochthonous” source of micronutrients, exerted by live cells together with organic polymers or taken up via absorption of ions from surrounding seawater. We added this information in the revised Conclusions.

Round 2

Reviewer 2 Report

I thank the authors for considering my comments and replying them with caution. My main concern with this manuscript was that it was based only on two samples, which is not enough to a robust statistical assessment. In their reply, the authors explained that the time and spatial variability in the composition of the macroaggregates during a particular event is fairly reduced according to the reports published; consequently, the samples analysed are enough to describe the characteristics of the particular event sampled. I do not doubt about that; however, I think that this reduced amount of samples does not permit assess statistically if the differences found between the three samples were due to two different events were sampled (2000 and 2004) or surface and water column were sampled (they are two possible variability sources and only two samples), even by assuming that the variability between 2000 and 2004 was reduced (how much is reduced?). Consequently, it is obvious that the sampling design is not suitable to test the hypothesis that “ 1) elemental composition of macroaggregates will be different depending on their location, i.e., floating on the water surface or suspended within the water column”. Testing this hypothesis will require, at least, analysing one sample from surface and from the water column during each event. In the new version, the authors presented results of a U-test that. I guess that they were calculated by using the three subsamples analysed from each sample. Please, note that the deviations among the three sub-samples are a measurement of the reliability of the method, not of the natural variability of the measured variables. Therefore, the results of the U-test cannot be used to assess differences between surface and water column or between 2000 and 2004 (independent samples should be used).  

Honestly, I cannot recommend the publication of a manuscript based on a so small amount of samples, even although the results are valuable. I encourage the authors to complete their analyses with additional samples as the work is really interesting.

Author Response

The main limitation of the present stud is rather small number of analyzed mucilage samples. However, based on numerous measurements of our group on mucilage aggregates of the Adriatic Sea using other techniques (e.g., refs. 1-4, 13-16, 20-21), we are confident that the collected samples (1) are representative for the mucilage event of specific month and year, and (2) spatial variations in other parameters of mucilage composition are not highly pronounced, provided that we clearly distinguish the surface and water column aggregates. Indeed, it is known that the horizontal and vertical distribution and accumulation of macroaggregates is heterogeneous and time-dependent on different factors such as the conditions in the water-column, the size, form and composition of the aggregates and the environmental conditions. However, the same (typical) characteristic biological and chemical composition of northern Adriatic macroaggregates were observed during every such events that have been largely described and confirmed by various authors [1,9,12,14,18, 27,53]. More specifically, the variations of chemical composition of macroaggregates are rather low, with a standard deviation rarely exceeding a few percent of the average, in terms of macro composition. This is illustrated by a systematic study of mean organic composition of macroaggregates collected over several seasons in the Pesaro region of the Adriatic Sea (i.e., ref. 14,53]. These authors demonstrated very low (i.e., ≤ 0.7 %) inter-annual variations of aggregates composition. Further, they also demonstrated rather stable chemical composition of both seawater and macroaggregates sampled in the Gulf of Trieste over the spring – summer –autumn seasons in 2004, with typical variations not exceeding ± 30%. Therefore, we believe that the selected samples are representative for the mucilage event of a specific month and year, and in general for Northern Adriatic macroaggregates. At the same time, we have to admit that detailed, spatially-resolved sampling of mucilage events across seasons for multi-elemental analyses is certainly necessary but it represents sizable, if not cost-prohibitive, research efforts that go beyond the scope of the present study.

Our second line of arguments to the reviewer is that we have used large volume (typically 1 L) of mucilage collected in the seawater. This represents large bulked sample which is equivalent of multiple subsamples taken in various places and as such it is sufficiently representative for mucilage events in the Adriatic Sea. We also added this information in the revised Methods description (revised section 2.2).

Reviewer 4 Report

The authors have adequately and thoroughly revised the manuscript in response to my comments. I can therefore recommend publication in its present form.

Author Response

We thank the reviewer for positive evaluation of our work and we did our best to further improve the manuscript during 2nd round of revision, via taking into account the comments of Editor and other reviewers

Reviewer 5 Report

Thank you very much to the authors for their hard work. The manuscript have been improved significantly. I am convinced with the answers of my questions.

Author Response

We are grateful for evaluation of our efforts by the reviewer. Thanks to new comments received from another reviewer and Academic Editor, we could further improve our manuscript during the 2nd stage of revision.